# Acid-Mediated Formation of Soybean Isolate Protein Emulsion Gels with Soybean Oil as an Active Component

**DOI:** 10.3390/foods12091754

**Published:** 2023-04-23

**Authors:** Chonghao Bi, Tong Zhou, Zeyuan Wu, Zhigang Huang

**Affiliations:** 1School of Artificial Intelligence, Beijing Technology and Business University, No. 11 Fu Cheng Road Haidian District, Beijing 100048, China; 2Beijing Key Laboratory of Quality Evaluation Technology for Hygiene and Safety of Plastics, Beijing 100048, China

**Keywords:** rheological properties, microstructure, analysis of fractals

## Abstract

In this study, the effect of soybean oil concentration on the rheology, water-holding capacity, and thermal stability of acid-mediated soy protein isolate (SPI) emulsion gels was investigated. The microstructure was analyzed and interpreted by CLSM and SEM observations. The results showed that the addition of soybean oil improved the elastic properties of the acid-mediated SPI emulsion gels. The storage modulus increased from 330 Pa (2% soybean oil concentration) to 545 Pa (8% soybean oil concentration) with a significant increase (*p* < 0.05). The increase in soybean oil concentration resulted in more SPI-coated oil droplets acting as active particles, enhancing the gel network. The acid-mediated SPI emulsion gels became more disordered as the soybean oil concentration increased, with the fractal dimension increasing from 2.92 (2%) to 2.95 (8%). The rheological properties, thermal analysis, and microstructure of 6% SPI gel and acid-mediated SPI emulsion gels with 2% to 8% soybean oil concentration were compared. The acid-mediated SPI emulsion gels with soybean oil as the active filler showed improved gel properties, greater thermal stability, and a homogeneous network structure compared to the acid-mediated SPI emulsion gels.

## 1. Introduction

Soybean is a major source of high-quality vegetable protein and is widely available and inexpensive [1]. The defatted soybean is transformed into soybean protein isolate by alkali extraction and acid precipitation. Soybean protein isolate is rich in the amino acids required by the human body and is beneficial to human health. This has led to the recognition of soybean protein isolate and its related food products by the general public [2]. The emulsion gelation properties of soybean protein are currently used in a wide range of applications in the food industry. Emulsion gels are complex colloidal materials that consist of a gel matrix with embedded droplets or a network formed from flocculated droplets [3,4]. Based on the state of the emulsion droplets, emulsion gels can be classified into two categories: emulsion-droplet-filled gels and emulsion-droplet-aggregated gels. Compared to conventional emulsions, emulsion gels have better storage stability because of their unique gel-like network structure and solid-like mechanical properties [5]. Recently, emulsion gels using proteins as emulsifiers have become a popular research topic due to their toughness and amphiphilic nature [6].

In recent years, many studies have investigated the physicochemical properties of emulsion-droplet-filled gels. For instance, Feng et al. (2019) studied the effect of Tween-20-stabilized emulsion concentration on beet pectin (SBP)/SPI and found that an emulsion concentration of 10 wt% yielded the highest hardness, adhesion, and chewiness, whereas an increase in emulsion concentration facilitated the improvement in gel swelling [7]. Mantovani et al. (2016) enhanced the gel networks of pretreated WPI emulsion gels by subjecting them to high-pressure homogenization, resulting in changes in the volume fraction of the dispersed phase and the adsorbed protein content [8]. Wang et al. (2016) proposed a poly (poppy oleic acid) and GDL-mediated protein gel prepared with glycerol, which does not require high-temperature treatment [9]. Liu et al. (2015) effectively improved the gel properties by heating SPI gels under extremely acidic (pH = 1.5) conditions for a long time (5 h) at low temperature (50 or 60 °C) to change the secondary and tertiary structures of proteins [10]. Zhao et al. (2020) reported an increase in the elastic modulus G′ of calcium-sulfate-mediated SPI gels and a gradual increase in gel hardness and water-holding capacity with the addition of soybean oil and an increase in soybean oil concentration [11].

Different matrix structures can be formed for protein-based emulsion gels using different preparation methods. Different gelation mechanisms and molecular forces between protein molecules in the gel matrix can result in different mechanical properties. For thermally mediated emulsion gels, non-covalent cross-linking (i.e., electrostatic interactions, hydrophobic interactions, and hydrogen bonding) and intermolecular disulfide bonds are the primary forces between spherical protein molecules [12]. TGase-mediated emulsion gels have more covalent cross-linking (i.e., ε-(γ-glutamyl)-lysine (G-L) cross-linking). Additionally, process parameters such as pH [13], temperature [14], protein content [15], and the addition of other components [16,17] can affect the structural and mechanical properties of protein-based emulsion gels. The droplet structure can also impact the mechanical properties of protein-based emulsion gels [5].

While current studies on the droplet structure of protein-based emulsion gels have focused on the emulsifier type, relatively little research has been conducted on the oil phase in acid-mediated emulsion gels of soybean isolate protein. Therefore, this study aims to investigate the effect of different soybean oil contents on acid-mediated emulsion gels of soybean isolate protein, including their rheological properties, thermal properties, microstructure, and fractal analysis. The research on this topic contributes to the further understanding of the structure of protein emulsion gels and improves the potential applications of soy protein in the food industry. At the same time, the subject helps to promote the development of emulsion gels as a substitute for animal fats, for example, in the delivery of bioactive substances.

## 2. Materials and Methods

### 2.1. Materials

Soybean protein isolate (dispersed, protein > 90%) and the glucono-δ-lactone (BR grade) were provided by Shanghai Macklin Biochemical Co., Shanghai, China; soybean oil (food grade) was purchased from Shenzhen Nanrui Grain and Oil Foodstuff Co., Shenzhen, China; Rhodamine B was purchased from Shanghai Macklin Biochemical Co., Shanghai, China.

### 2.2. Sample Preparation

#### 2.2.1. SPI emulsion for Gel Formation

For the preparation of the SPI emulsion for gel formation, a 6% (*w/v*) stock solution of SPI was prepared by stirring the SPI powder in deionized water (pH = 7.0 ± 0.3) at 25 °C for 30 min. The solution was then heated to 90 °C for 30 min and cooled to 25 °C in cold water. Five stock solutions containing 0%, 2%, 4%, 6%, and 8% soybean oil were prepared by adding soybean oil to the SPI stock solution and subjecting it to high shear for 1 min at 5000 rpm using a high-speed shear disperser (IKA T25; IKA Instruments, Staufen im Breisgau, Germany). The resulting mixtures were subjected to high-pressure homogenization (APV-2000; Hammertech Fluid Technology Co., Beijing, China) at 500 bar and stored in sealed storage at 4 °C overnight.

#### 2.2.2. Acid-Mediated SPI Emulsion Gels

For the acid-mediated SPI emulsion gels, 10 mL of different concentrations of SPI–soybean oil mixture (0%, 2%, 4%, 6%, and 8%) were taken and glucono-δ-lactone (GDL) powder was added to each (GDL conc. = 2% *w/v*) and placed at room temperature for 1 h. The resulting mixtures were stored at 4 °C for future use.

#### 2.2.3. Acid-Mediated Emulsion Gels for CLSM Test

The experiment was carried out by confocal microscopy to observe the microstructure of the gel. The SPI–soybean oil mixed solutions were dyed with 0.01% Rhodamine B agent and 2% GDL was added. The stained samples were left at room temperature for one hour to form the gel.

#### 2.2.4. Freeze-Drying Acid-Mediated SPI Emulsion Gels

For the freeze-drying of the SPI gel and acid-mediated SPI emulsion gels, the samples were placed in a freeze dryer and pre-frozen in a chamber at −60 °C. The frozen samples were then placed in a closed vacuum vessel for heating to dehydrate and dry the samples.

### 2.3. Rheological Test

#### 2.3.1. Gel Formation

The samples were cast onto a Peltier plate with a diameter of 40 mm and a gap of 1 mm for rheological studies. The DHR-2 rheometer (TA Instruments Ltd., Crawley, UK) was used for all rheological measurements, including time scan tests, frequency scan tests, and creep recovery tests. The acid-mediated SPI emulsion gels were loaded with a standby temperature of 5 °C for the Peltier paste plate, which was connected to a water circulation pump for temperature control. To prevent moisture from evaporating from the sample, silicone oil was added to the top and around the plate. Once the temperature of the sample was stabilized at 20 °C for 5 min, dynamic rheological testing could be performed. Gluconolactone (GDL) was added to bring the pH of the sample down to 4.5 and form gels when the temperature of the plate was increased from 20 °C to 60 °C. Data were recorded throughout the experiment, and three parallel experiments were conducted per sample.

#### 2.3.2. Time Sweep Test

During the gel formation process, time scan tests were conducted at a constant frequency of 1 Hz and a temperature of 60 °C, where the energy storage modulus (G′) was recorded. To ensure that the strain amplitude of all samples remained within the linear viscoelastic region, an angular frequency of 6.283 rad/s and a strain amplitude of 1% were selected based on the strain scan test. The changes in dynamic modulus (G′: storage modulus; G″: loss modulus) of the hybrid system were monitored over time. The energy storage modulus G′ of the acid-mediated SPI emulsion gel was found to follow a kinetic equation as a function of time (*t*):*G*′(*t*) = *G*′_∞_ [1 − *e*^−*K*(*t*−*t*_*g*_)^](1)
where *t* (s) is time, *t_g_* (s) is the time of achieving gelation, *K* (1/s) is the gel rate constant, and *G*′*_∞_* (Pa) is the estimation of *G*′ (Pa) at infinity time [18].

#### 2.3.3. Frequency Sweep Test

The energy storage modulus G′ and loss modulus G″ of the gel were measured while maintaining the strain value at 1%, within the angular frequency range of 0.1 rad/s to 10 rad/s. The correlation between G′ (Pa) and G″ (Pa) with angular frequency *ω* (rad) was determined using a power-law equation, as follows:*G*′ = *K*′**ω*^*n*^′(2)
*G*″ = *K*″**ω*^*n*^″(3)
where *K′* (Pa*sn/rad) and *K*″ (Pa*sn/rad) denote the power law constants; *n*′ (dimensionless) and *n*″ (dimensionless) denote the frequency index; the value indicates the storage modulus *G*′ (Pa) or loss modulus *G*″ (Pa) and the dependence on frequency [19].

#### 2.3.4. Creep/Recovery Test

At a temperature of 20 °C, the gel samples were subjected to a transient stress of 7.958 Pa, and the resulting strain (%) and flexibility J (Pa-1) were measured for 120 s during the creep phase. When the creep phase was complete, the stress was instantaneously reduced to 0. Testing of the sample continued for a duration of 120 s. The creep curve of the acid-mediated SPI latex was divided into three phases, as represented by the following equation:(4)$$\frac{\gamma }{\sigma }=\frac{\gamma_{H}}{\sigma }+\frac{\gamma_{R}}{\sigma }+\frac{\gamma_{N}}{\sigma }$$
where *γ* (%)—total strain; *γ_H_* (%)—strains associated with elastic; *γ_R_* (%)—strains associated with retarded elastic; γ*_N_* (%)—strains associated with irreversible deformations.

The experimentally derived creep curves were fitted using the four-element Maxwell–Voigt model, which is given by Equation (5):(5)$$J\left(t\right)=\frac{1}{G_{H}}+\frac{1}{G_{V}}\left(1-e^{-\frac{t}{\tau }}\right)+\frac{1}{\eta_{N}}$$
(6)$$\tau =\frac{\eta_{V}}{G_{V}}$$
where *J*(Pa−1)—compliance (ratio of strain to stress); *G_H_* (Pa)—elastic modulus (Hooke element); *G_V_* (Pa)—elastic modulus (Voigt element); *Τ* (s)—retardation time; *η_V_* (Pa s)—viscous component (Voigt element); *η_N_* (Pa s)—viscous component (Newtonian element).

Recovery rate *R* (%):(7)$$R\left(\%\right)=\frac{\gamma_{T}-{\;\gamma }_{P}}{\gamma_{T}}\;\times \;100$$
where *γ_T_* (%)—total strain; *γ_P_* (%)—recovery strain [18].

### 2.4. CLSM Test and Analysis of Fractals

To prepare the samples for microscopic examination, the method described in Section 2.2.3 was used. Specifically, 0.5% rhodamine B was added to 10 mL of the mixture and the samples were then separately treated with 2% gluconolactone powder, stirred, and sampled. The microstructure of acid-mediated SPI emulsion gels was observed and imaged using one of the confocal laser scanning microscopes (CLSMs) in a LEICA TCS SP5 II. The He-Ne/visible laser source in the CLSM emits a 543 nm laser through the sample, where the rhodamine B in the sample is excited by this laser. Emission wavelengths ranged from 561 to 700 nm. Imaging is observed through an inverted microscope in the CLSM to obtain an image picture [20]. The method of Bi et al. (2013) was referenced with minor changes. The fractal parameters of the acid-mediated SPI latex were calculated by CLSM image analysis [21]. Specifically, the RGB image of 1024 × 1024 pixels was converted into an 8-bit gray image and then subjected to black and white binarization using the middle value of the gray level histogram of each image as a threshold. The box-counting method was used to calculate the fractal dimension Df (dimensionless) of the acid-mediated SPI emulsion gel samples, based on the scaling model given by Equations (8) and (9):(8)$$D_{P}=-\frac{\log N_{\varepsilon }}{\log\varepsilon }$$
(9)$$D_{f}=D_{P\;}+1$$

### 2.5. Electron Microscope Scanning Test (SEM)

The microstructure of the acid-mediated SPI-emulsion gels with varying soybean oil concentrations was studied using scanning electron microscopy (SU1510, Hitachi, Japan). The samples were freeze-dried to create a mixed powder, and a layer of gold was sputter-coated using an SCD-005 ion sputter. The fracture morphology of each sample was observed at an accelerating voltage of 5.0 kV and a magnification of ×20,000 in the SEM chamber [22].

### 2.6. Thermal Analysis

An amount of 0.2 g of GDL was added to 10 mL of sample and stirred to homogenize. The mixed sample was heated in a water bath at 60 °C for 20 min to form a gel. The gel was allowed to cool and then freeze-dried to make an SPI gel powder for DSC testing. The DSC test was carried out by increasing the temperature from 40 °C to 200 °C at a rate of 20 °C/min and recording the change in heat flow with temperature during the temperature rise and fall. This was used to analyze the thermal stability of the material [23].

In addition, the thermogravimetric stability of the samples was determined using a TGA instrument (TA Q50, Waters Corporation, Milford, MA, USA). A 5 mg sample (in powder form) was placed in a platinum cup and the temperature was increased from 40 °C to 500 °C in a nitrogen atmosphere by a ramping rate of 50 °C/min. Weight changes were recorded, and the TGA and DTG curves were calculated using TRIOS software. Heat changes during heating and cooling were also recorded. Three replicates were run for each sample [24].

### 2.7. Water-Holding Gel Test (WHC)

The water-holding capacity of the samples was determined using the centrifugation method. The gel samples were placed in a centrifuge tube at room temperature and centrifuged for 20 min. After centrifugation, the supernatant was poured off to remove residual water. The remaining sample was weighed. Equation (10) was used to calculate the WHC of the gel:(10)$$\mathrm{WHC}\left(\%\right)=\frac{M_{t}-{\;M}_{w}}{M_{t}}\;\times \;100\%$$
where *M_t_* (g)—weight of the gel before centrifugation; *M_w_* (g)—weight of water released from the gel after centrifugation [25].

### 2.8. Data Processing and Statistical Analysis

The box-counting method was conducted using the open-source software Image J 1.44p (National Institutes of Health, Bethesda, MD, USA). Three parallel experiments were performed for each experiment. Results are presented as mean ± standard deviation. Data were processed using one-way ANOVA and Duncan’s test in the SPSS statistical package (SPSS 16.0, IBM, New York, NY, USA). The statistical significance between two means was determined at the 95% confidence level (*p* < 0.05). Regression equations were fitted and analyzed using the SPSS packagej.

## 3. Result and Discussion

### 3.1. Time Sweep Test

The trend of the energy storage modulus G′ and G″ with time during gel formation at different oil concentrations is depicted in Figure 1A. The rheological properties of the emulsion gel system depend on the continuum term, and a protein-based emulsion was formed due to the small volume fraction of oil. Until 120 s, G′ was approximately equal to 0 for all samples, indicating that the gel was not formed at this stage (Bi et al., 2013) [21]. As the disaggregation rate of GDL accelerated, G′ showed an increasing trend and then reached a steady state around 600–700 s. The elastic properties of the samples were increasing during this time, and the elastic properties of the gels were stronger with higher oil concentrations. This was due to the fact that the soybean-isolate-coated oil droplets behaved as active filler particles and had a high modulus relative to the matrix, allowing the samples to obtain structural enhancement. A similar phenomenon was observed in the study by Dille et al. (2015) [26].

The effect of oil concentration on the G′*_∞_*, K, and tg of the mixed gels is displayed in Table 1A. G′*_∞_* increased significantly (*p* < 0.05) when the oil concentration increased, and the estimated maximum value that the energy storage modulus G′ may reach increased significantly (*p* < 0.05). This result indicates that acid-mediated SPI emulsion gels produced more solid gel structures compared to SPI gels. This may be due to the particle–gel network interaction formed between SPI and soybean oil [26]. tg values decreased from 152.565 s to 141.003 s, with an earlier time for the onset of gelling formation. This observation suggests that the addition of soybean oil can promote the start of gel formation. The value of K increased from 0.004 to 0.005 with the addition of 2% soybean oil, but the value of K remained constant with the further increase in oil concentration. This indicates that the oil phase emulsified and then formed a gel faster after the addition of soybean oil.

Overall, the data of G′*_∞_*, K, and tg suggest that the presence of soybean oil accelerates the formation procedure of mixed gels and also enhances the elastic properties of the gels.

### 3.2. Frequency Sweep

Figure 1B,C illustrate the variation in G′ and G″ with frequency from 10 rad/s to 100 rad/s for gels with different oil concentrations. It is apparent that for all emulsion gels with oil concentration, G′ was greater than G″, indicating solid-like properties. A significant increase (*p* < 0.05) in G′ was observed for the acid-mediated SPI emulsion gels with 6% and 8% oil concentrations, suggesting that a turning point in their performance may lie between these two concentrations.

Table 1B indicates that the n′ values of the acid-mediated SPI emulsion gels with 2% soybean oil concentration were not significantly higher (*p* < 0.05) than those of SPI gels, and the soybean oil concentration peaked at around 4%. The analysis suggests that increasing the soybean oil concentration to 4% enhanced the gel properties of the mixed gels, but the n′ value increased significantly (*p* < 0.05) beyond 4% and decreased instead with the further increase in soybean oil concentration. Therefore, soybean oil concentration plays a vital role in improving the fluid properties of the acid-mediated SPI emulsion gels [22].

As per Tang et al. (2019), the frequency dependence of the elastic properties of acid-mediated SPI emulsion gels reduced their frequency stability [27]. The regression coefficients for all samples were greater than 95%, indicating a good fit for all samples.

### 3.3. Creep Recovery

The findings from Figure 2 show that the strain rate gradually decreased until 60 s and then smoothed out with a linear relationship between strain and time from 60 s to 100 s. This phenomenon indicates an increase in the effect of viscous flow on sample deformation. The deformation in the recovery phase is due to viscous flow, which is the viscous part of the total strain (*γ_T_*) [28]. The total deformation of the acid-mediated SPI emulsion gels decreased with increasing soybean oil concentration, and the magnitude of the total deformation can be used to compare the stiffness of the gel samples. The addition of soybean oil resulted in a significant increase (*p* < 0.05) in the structural stiffness of the acid-mediated SPI emulsion gels, indicating that soybean oil acted as a filler in the cross-linking process and helped to form a denser network structure [29].

The creep rate decreased with increasing soybean oil concentration, indicating a greater contribution of the viscous component to creep when soybean oil was added to SPI to produce acid-mediated SPI emulsion gels. The amount of flexibility J also showed a decreasing trend with increasing soybean oil concentration, suggesting the formation of more rigid hybrid gels. The soybean-isolate-protein-coated oil droplets behaved as active filler particles with a higher modulus relative to the matrix, allowing the samples to acquire structural enhancement.

Table 2A lists the regression parameters obtained by fitting the Maxwell–Voigt model to the experimental data, which shows that the model can fit the experimental data well (*R^2^* ≥ 0.995) [18]. The GH and GV values increased with higher soybean oil concentrations, indicating higher or rapid transient elastic behavior in SPI gels. The *η_V_* values were related to the stiffness and viscous orientation of the amorphous polymer chains over a short period of time, and they also increased with higher soybean oil concentrations. These trends suggest that soybean oil acted as a filler in the cross-linking, helping to form a denser network structure. This conclusion was consistent with the results of the thixotropic experiments.

The acid-mediated SPI emulsion gel enhanced the orientation of the gel matrix and increased the probability of connections between molecular chains, resulting in a stronger network structure. The retardation time *τ* did not differ significantly (*p* < 0.05) between samples. The acid-mediated SPI emulsion gel recovery rate decreased by adding soybean oil of 2% concentration compared to the SPI gel, and thereafter, it increased with increasing soybean oil concentration, with the highest recovery rate of acid-mediated SPI emulsion gels being 92.20% at a soybean oil concentration of 8% (Table 2B).

### 3.4. Microstructure and Fractal Dimension of Acid-Mediated SPI Emulsion Gels

The microstructures of 6% SPI gels and acid-mediated SPI emulsion gels with different soybean oil concentrations (2%, 4%, 6%, 8%) were examined using CLSM images, as shown in Figure 3A–E. The SPI network was stained with Rhodamine B, appearing as red (bright) areas, while the oil fraction appeared as dark areas. The images indicated that the microstructures of the two types of gels had different textures and porosity. The distribution of soybean oil in the gel when used as an active ingredient was also visually explained. Similar findings were reported by Guo et al. (2022), who observed that soybean oil droplets were small and uniformly surrounded by water when the soybean oil concentration was 2%, but gradually accumulated into clusters as the concentration increased [30]. A positive correlation was observed between the void space of the oil droplet volume protein network and the concentration of soybean oil.

The fractal dimension (Df) was used to characterize the complexity of the gel network, as reported by Bi et al. (2014) [29]. The Df values of 6% SPI gels and acid-mediated SPI emulsion gels with soybean oil concentrations of 2%, 4%, 6%, and 8% are presented in Figure 3a–e. The acid-mediated SPI emulsion gels with 8% soybean oil content exhibited the highest Df value of 2.9422, and the Df value increased with the increase in soybean oil concentration. This indicated that the addition of soybean oil increased the disorganization of the complex network structure of the SPI emulsion gels, and the acid-mediated complex network structure became more disorganized with the increase in soybean oil concentration.

SEM images of SPI gels and acid-mediated SPI emulsion gels with 8% oil concentration are shown in Figure 3F,G. The SPI gel network was found to be inhomogeneous with small aggregation, loose structure, and irregular pores, while the addition of soybean oil resulted in a denser and more homogeneous gel network with smaller pores. This was attributed to the interfering effect of protein microgels between the oil droplets, effectively preventing flocculation and agglomeration and resulting in stable emulsions. Similar findings were reported by Guo et al. (2022) and Chu et al. (2022) [30,31]. The SEM results were consistent with the properties of the gels obtained from rheological tests.

### 3.5. Thermal Analysis

According to the results shown in Figure 4A, the DSC analysis of the acid-mediated SPI emulsion gels revealed two absorption peaks in the thermal characteristic curve. The first peak was observed when the temperature reached 90 °C (80~110 °C) and represented the denaturation temperature of the SPI protein. Specifically, the SPI denatured at around 90 °C. The second absorption peak occurred at 160 °C and was attributed to the absorption peak temperature of soybean oil due to its presence in the mixed system. This phenomenon is consistent with a study conducted by Bi et al. (2020) [32]. Table 3A indicated that, as the concentration of soybean oil increased, the enthalpy value corresponding to the first absorption peak decreased during the process of temperature rise. This could be explained by the hydrophobic groups in the protein binding to the lipid molecules, causing the lipid molecules to be ordered and covered in the protein gel network. As a result, a stable protein–liposome system was formed, resulting in a less exothermic sample during the warming process and an increased thermal stability of the sample. Moreover, as the concentration of soybean oil increased, the minimum absorption peak temperature of the acid-mediated SPI emulsion gels increased with increasing oil concentration until it peaked at 4% oil concentration, then decreased, and finally increased again as the oil concentration continued to increase. This was likely due to the fact that the SPI-coated oil droplets acted as reactive packing particles until the oil concentration reached 4%, and their interaction with the matrix increased with increasing oil concentration, resulting in more heat being required for full denaturation of the soybean protein. However, at oil concentrations above 4%, soybean-isolate-protein-coated oil droplets behaved as inactive packing particles, reducing the temperature required for the complete denaturation of soy protein that was not denatured during gel formation.

Figure 4B shows the thermogravimetric curves of the acid-mediated SPI emulsion gels with different concentrations of soybean oil. It was observed that the temperature of the acid-mediated SPI emulsion gel at 50% weight loss increased from 350 °C (2%) to 420 °C (8%) with the increase in soybean oil concentration. The thermal degradation of SPI gels and acid-mediated SPI emulsion gels between 100 °C and 500 °C occurred in three steps. The first step before 150 °C was due to the evaporation of the water absorbed in the gel. The second step between 170 °C and 250 °C was caused by the evaporation of soybean oil and further curing reactions of the gel, generating gas. The mass loss occurring between 270 °C and 450 °C was mainly due to the degradation of the cross-linked network structure. As the concentration of soybean oil increased, the position of the first peak moved in the direction of increasing temperature, indicating that the temperature required for further curing of the acid-mediated SPI emulsion gel increased with increasing soybean oil concentration. The second peak moved in the direction of increasing temperature, indicating that the thermal stability of the gel was increased by the addition of soybean oil, which altered the original structure. This finding is consistent with the study by Bi et al. (2020) [32].

Figure 4C and Table 3B revealed that the rate of weight loss of the acid-mediated SPI emulsion gels changed with increasing temperature T and had three distinct upward absorption peaks. The temperature of all three peaks increased with increasing soybean oil concentration, indicating that the thermal stability of the acid-mediated SPI emulsion gels increased with increasing soybean oil concentration. The second absorption peak corresponded to the largest rate of weight loss, indicating that the samples decomposed at the fastest rate at the second absorption peak. As the concentration of soybean oil was raised, the weight loss rate of the first and third absorption peaks decreased, while that of the second absorption peak increased. This indicates that the addition of soybean oil can heighten the temperature required for protein denaturation within the gel, thereby augmenting the thermal stability of the gel. The findings suggest that some extent of intermolecular interaction between soybean isolate and soybean oil might have taken place, leading to the formation of robust cross-linking and a sturdy network structure, which is beneficial for enhancing the thermal stability of the gels.

### 3.6. Water-Holding Capacity

Figure 5 depicts the water-holding capacity of soybean isolate protein (SPI) gels and acid-mediated SPI emulsion gels with varying soybean oil concentrations at centrifugation speeds of 500 r/min and 1000 r/min. The results showed that the addition of soybean oil slightly increased the water-holding capacity of the SPI gels, while the two were proportional to each other. This may be due to the fact that as the soybean oil concentration increased, more soybean-isolate-coated oil droplets acted as active particles, strengthening the gel network and enhancing water retention [33]. The emulsion gel matrix more effectively bound water, resulting in a more uniform and dense network structure, which improved the water retention of the gel. These findings are consistent with the results of Ma et al.’s study (2022) [34].

Additionally, for gels formed at the same soybean oil concentration, their water retention at different rotational speeds varied, with the water retention of acid-mediated SPI emulsion gels decreasing as the rotational speed increased. This may be due to the increase in centrifugal speed, which breaks down the active particles, resulting in protein aggregation and an increase in particle size. This loosens the network structure, making it easier for water molecules to be released, ultimately reducing the water retention of the final gel [35].

## 4. Conclusions

The study investigated how different concentrations of soybean oil affected acid-mediated SPI emulsion gels in terms of gel rheology, thermodynamics, microstructure, and water retention. The results indicated that the addition of soybean oil accelerated the formation of the emulsion gels and improved their properties significantly with increasing concentration of soybean oil. The microstructure of the gels became more homogeneous with the addition of soybean oil, but also more disorganized in their complex network structure. The concentration of soybean oil increased the number of SPI-coated oil droplets as active particles, which improved the gel network and resulted in a stronger cross-linking between SPI and soybean oil. The water-holding capacity of the gel also increased significantly with increasing soybean oil concentration. Furthermore, a certain concentration of soybean oil (less than 4%) improved the thermal stability of the acid-mediated SPI emulsion gels. Overall, these findings have important implications for the utilization of soybean isolate and the development of functional foods.

## Figures and Tables

**Figure 1 foods-12-01754-f001:**
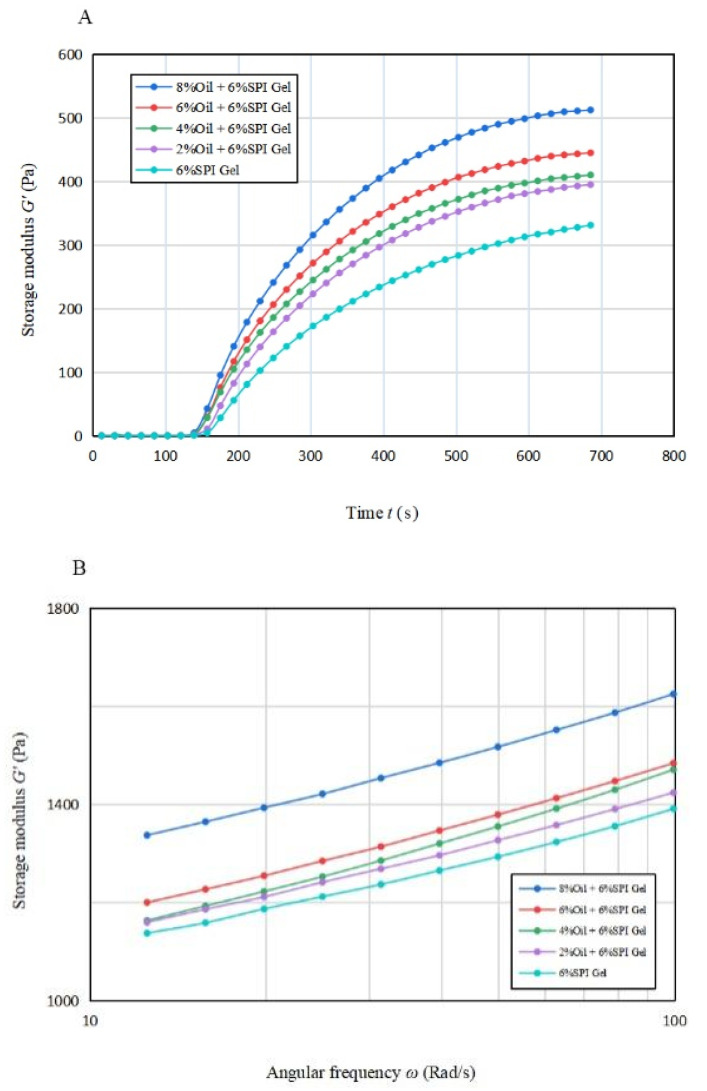
(**A**) The variation in energy storage modulus with time for acid-induced SPI emulsion gels with different soybean oil concentrations; (**B**,**C**) The variation in energy storage modulus and loss modulus with angular frequency for acid-induced SPI emulsion gels with different soybean oil concentrations.

**Figure 2 foods-12-01754-f002:**
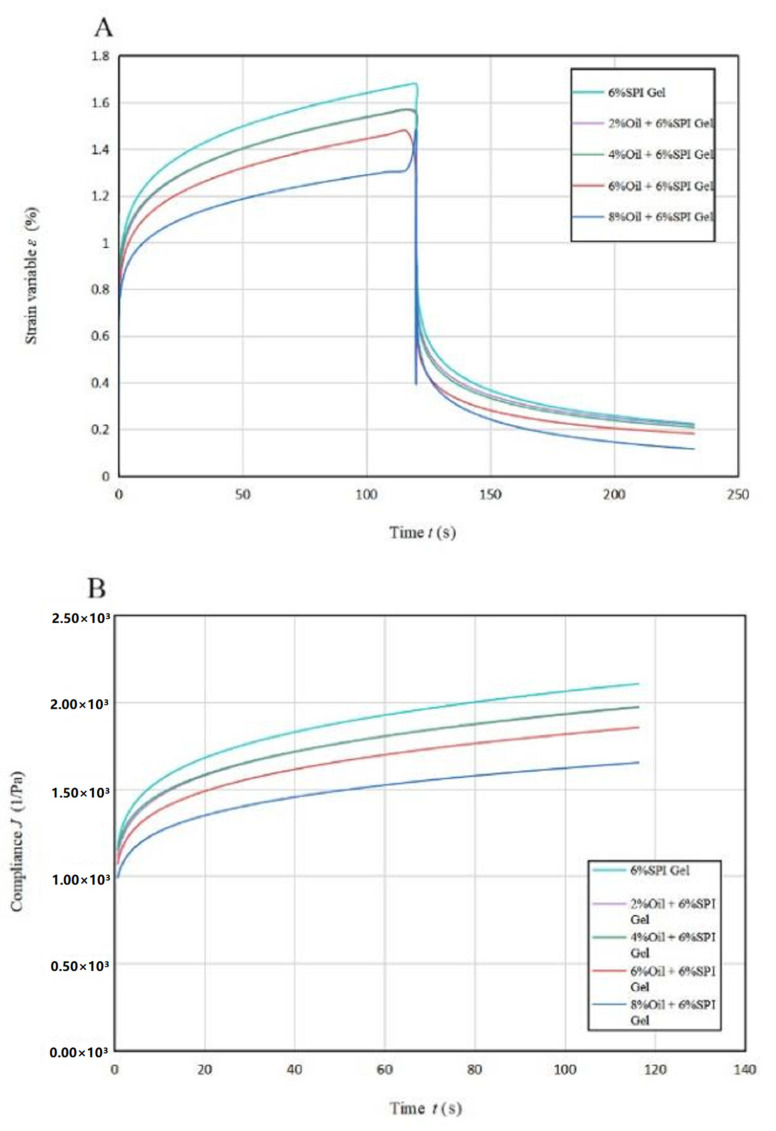
Creep recovery curves (**A**) and creep stress curves (**B**) for acid-induced SPI emulsions at different oil concentrations.

**Figure 3 foods-12-01754-f003:**
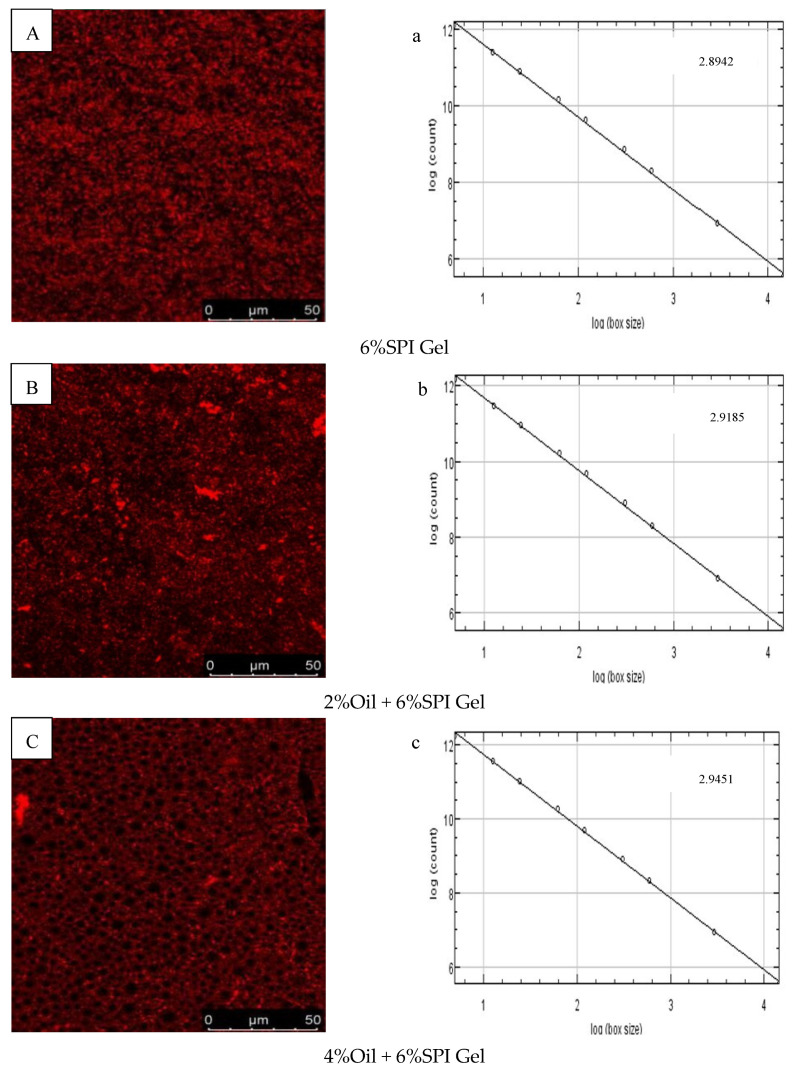
(**A**–**E**) CLSM images of different-oil-concentration acid-induced SPI emulsion gels; (**a**–**e**) analysis of fractals of gels; (**F**,**G**) scanning electron microscopy microstructures of gels.

**Figure 4 foods-12-01754-f004:**
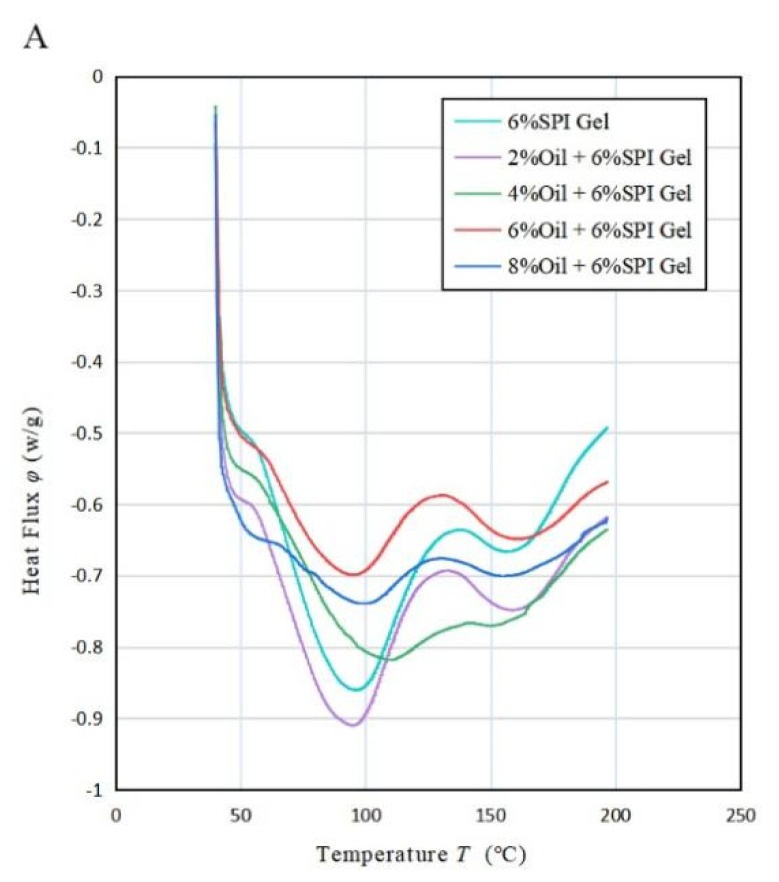
(**A**) Variation in heat flow rate with temperature *T* for the mixed emulsion of SPI and soybean oil at a heating rate of 20 °C/min, (**B**) variation in residual mass percentage with *T* for the mixed emulsion at 50 °C/min, (**C**) variation in weight loss change rate with temperature *T* for the mixed emulsion at 50 °C/min.

**Figure 5 foods-12-01754-f005:**
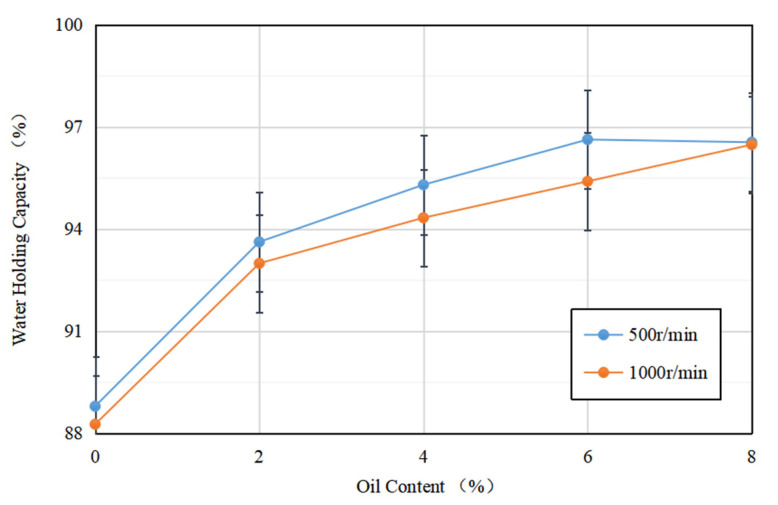
Variation in water-holding capacity of acid-induced SPI emulsion gels with soybean oil concentration at 500 r/min and 1000 r/min centrifugation speed.

**Table 1 foods-12-01754-t001:** (**A**) Effect of different oil concentrations on each parameter (*G*′*_∞_*, *K*, *t_g_*) of acid-induced SPI emulsion gels. (**B**) Effect of soybean oil concentration on acid-induced SPI emulsion gel frequency scan power law constants *K*′, *K*″ and frequency indices *n*′, *n*″.

(**A**)						
**Oil** **Concentration**	**G′_∞_**		**K**	**t_g_**		**R^2^**
**%, *v*/*v***	**Pa**		**1/s**	**s**		**%**
0	374.153 ± 2.135 ^a^		0.004 ^a^	152.565 ± 0.764 ^b^		100
2	439.681 ± 1.418 ^b^		0.005 ^b^	149.822 ± 0.525 ^b^		100
4	439.055 ± 1.721 ^b^		0.005 ^bc^	143.174 ± 0.728 ^a^		100
6	472.587 ± 1.859 ^c^		0.005 ^c^	143.607 ± 0.752 ^a^		100
8	544.840 ± 2.097 ^d^		0.005 ^c^	141.003 ± 0.765 ^a^		100
(**B**)						
**Oil** **concentration**	**G′ = K′*ω^n^′**			**G′ = K″*ω^n^**″		
**%, *v/v***	**K′**	**n′**	**R^2^**	**K″**	**n″**	**R^2^**
0	892.139 ± 2.931 ^b^	0.096 ± 0.001 ^b^	99.9%	129.695 ± 1.680 ^c^	0.178 ± 0.003 ^b^	99.7%
2	911.285 ± 2.262 ^c^	0.096 ± 0.001 ^b^	99.9%	130.854 ± 1.970 ^c^	0.159 ± 0.004 ^a^	99.5%
4	876.587 ± 2.047 ^a^	0.112 ± 0.001 ^d^	99.9%	125.341 ± 1.549 ^a^	0.172 ± 0.003 ^b^	99.7%
6	929.127 ± 2.464 ^d^	0.101 ± 0.001 ^c^	99.9%	118.650 ± 1.411 ^a^	0.163 ± 0.003 ^a^	99.7%
8	1056.525 ± 2.570 ^e^	0.093 ± 0.001 ^a^	99.9%	131.465 ± 1.462 ^c^	0.163 ± 0.003 ^a^	99.6%

^a,b,c,d,e^ Values in a column with different superscripts were significantly different (*p* < 0.05).

**Table 2 foods-12-01754-t002:** (**A**) Maxwell–Voigt model parameters of acid-induced SPI emulsion gels as a function of soybean oil concentration. (**B**) Effect of soybean oil concentration on the recovery rate of acid-induced SPI emulsion gels.

(**A**)						
**Oil Concentration**	**G_H_**	**G_V_**	**τ**	**η_N_**	**η_V_**	**R^2^**
**%, *v*/*v***	**Pa**	**Pa**	**S**	**Pa*s**	**Pa*s**	**%**
0	861.635 ± 5.321 ^a^	2075.571 ± 42.104 ^a^	7.398 ± 0.386 ^a^	230,742.278 ± 7932.363 ^a^	15,355.074	99.5
2	903.273 ± 5.428 ^bc^	2321.979 ± 47.892 ^ab^	7.226 ± 0.385 ^a^	246,432.549 ± 8199.358 ^a^	16,778.620	99.5
4	939.652 ± 4.960 ^c^	2539.683 ± 51.159 ^b^	7.678 ± 0.394 ^a^	272,826.116 ± 9029.826 ^ab^	19,499.686	99.6
6	873.812 ± 4.507 ^ab^	2491.641 ± 50.563 ^b^	7.494 ± 0.390 ^a^	251,036.066 ± 7834.447 ^a^	18,672.358	99.6
8	1018.474 ± 5.192 ^d^	2983.496 ± 59.241 ^c^	7.191 ± 0.369 ^a^	319,116.026 ± 10,294.577 ^b^	21,454.320	99.6
(**B**)						
**Oil concentration**			**Recovery rate**			
**%, *v*/*v***			**%**			
0			86.64			
2			86.03			
4			86.70			
6			87.66			
8			92.20			

^a,b,c,d^ Values in a column with different superscripts were significantly different (*p* < 0.05).

**Table 3 foods-12-01754-t003:** (**A**) Effect of soybean oil concentration on the enthalpy and peak value corresponding to each exothermic peak in the DSC curve of acid-induced 6% (*w/w*) soybean isolate protein emulsion. (**B**) Variation in the weight loss ratio corresponding to the weight loss region of the TGA curve and the peak temperature of the DTG curve for 6% (*w/w*) SPI mixed with each concentration of soybean oil emulsion.

(**A**)							
**Oil** **Concentration**	**Weight**	**Protein Peak Min T_1_**	**Heat Flow 1**	**Enthalpy Value 1**	**Oil Peak Min T_2_**	**Heat Flow 2**	**Enthalpy Value 2**
**%, *w/w***	**mg**	**°C**	**w/g**	**J/g**	**℃**	**w/g**	**J/g**
0	2.507	95.930	−0.864	36.465	156.230	−0.043	6.741
2	3.289	95.905	−0.912	31.468	158.665	−0.749	7.832
4	4.509	110.115	−0.819	18.776	149.620	−0.771	1.791
6	2.597	95.080	−0.700	16.389	160.735	−0.649	7.282
8	2.386	98.550	−0.739	7.719	155.180	−0.701	4.896
(**B**)							
**Oil content**	**Peak value 1**	**Weightlessness ratio 1**	**Peak Value 2**	**Weightlessness Ratio 2**		**Peak value 3**	**Weightlessness ratio 3**
**%**	**°C**	**%**	**°C**	**%**		**°C**	**%**
0	201.64	16.77	321.09	32.38		N/A	33.72
2	210.95	11.10	326.57	29.67		380.65	35.82
4	211.68	11.02	326.46	48.47		395.82	20.48
6	210.61	9.64	335.11	53.08		401.53	16.28
8	215.11	7.68	331.60	54.95		414.89	16.25

## Data Availability

Data are contained within the article.

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
