# Peer review of "Acid-Mediated Formation of Soybean Isolate Protein Emulsion Gels with Soybean Oil as an Active Component"

_foods, 2023, doi:10.3390/foods12091754_

Round 1

Reviewer 1 Report

The manuscript has investigated the influence of soybean oil concentration on the acid-mediated Soy protein isolate emulsion gels and has reported interesting results. However the manuscript in its present form must be improved. 

Comments:

- Introduction: Please add more details about SPI and its functional properties. 

- All of the tables and figures are presented in the supplementary files . It is suggested to add some of the figures and tables to the main document. 

- In some of the tables you have used "a" for the highest value while in other tables "a" is used for the lowest value. 

- Statistical analysis in Table 1B is not correct. 

Reviewer 2 Report

It is a well designed study which brings interesting conclusion with a potential use in the food industry. The manuscript had been significantly improved, however I still find several aspects to be improved. 

it prevents cardiovascular disease” it is overestimation, please change this sentence.

Many reperations f.ex. “t is widely used in the food industry. The emulsion gelation properties of soy protein are currently used in a wide range of applications in the food industry

Tables. I think it is not necessary to put 3 numbers after dots.

Reviewer 3 Report

Manuscript ID: Foods 2023, 12, x. https://doi.org/10.3390/xxxxx

Title: Acid-mediated Formation of Soybean Isolate Protein Emulsion Gels with Soybean Oil as an Active Component

Authors: Chong-hao Bi, Tong Zhou, Ze-yuan Wu and Zhi-gang Huang

Overview and  general  recommendation: 

In this manuscript, the authors studied the effect of soybean oil concentration (0%, 2%, 4%, 6%, and 8%) on acid-mediated soy protein isolate (SPI) emulsion gels was investigated. Based on the Methodological section, the following analyses were performed rheological tests (gel formation, time sweep test, frequency sweep test, creep/recovery test), CLSM test and analysis of fractals, electron microscope scanning test (SEM), thermal analysis, water-holding gel test (WHC), and statistical analysis. According to the findings of this study, adding soybean oil affected the final emulsion gel. It improved the final gel structure's properties and made it more disorganized. Furthermore, the concentration of the added oil (below 4%) had a significant enhancing effect on the thermal stability of the acid-mediated emulsion gels. The manuscript's subject is quite interesting. The studies are thoroughly described and supported by literature.

Major comments:

·      The manuscripts lack tables and figures crucial to understand the results (why they are at "supplementary files').

·      The authors did not explain where such emulsion gel can be used in food applications, which is essential considering the journal's scope.

·      The abstract does not specifically showcase what analyses were performed.

·      The authors did not prove the presence of active fillers using CLSM or SEM. 

 Minor  comments:  

“2.7. Data processing and statistical analysis” - In my opinion the authors should more specifically describe what statistical tools/ test they performed on the acquired data.
